# Biodegradation and Cell Behavior of a Mg-Based Composite with Mesoporous Bioglass

**DOI:** 10.3390/ma16186248

**Published:** 2023-09-17

**Authors:** Yan Zhou, Dongsheng Wang, Youwen Yang

**Affiliations:** 1Key Laboratory of Construction Hydraulic Robots, Anhui Higher Education Institutes, Tongling University, Tongling 244061, China; zhouyan099@163.com; 2Institute of Additive Manufacturing, Jiangxi University of Science and Technology, Nanchang 330013, China

**Keywords:** biodegradable magnesium alloy, corrosion resistance, in situ deposition, Mg-based composite, mesoporous bioglass, osteogenesis

## Abstract

Biodegradable magnesium (Mg) and its alloys show tremendous potential as orthopedic materials. Nevertheless, the fast degradation and insufficient osteogenic properties hinder their applications. In this study, mesoporous bioglass (MBG) with an ordered branch-like structure was synthesized via a modified sol–gel method and showed a high specific surface area of 656.45 m^2^/g. A Mg-based composite was prepared by introducing the MBG into a Mg matrix via powder metallurgy. Degradation tests showed that the introduction of MBG increased the adsorption sites for Ca and P ions, thus promoting the formation of a Ca-P protective layer on the Mg matrix. The Ca-P protective layer became thick and dense with an increase in the immersion time, improving the protection ability of the Mg matrix, as proven by electrochemical impedance spectroscopy measurements. Meanwhile, the Mg-based composite also exhibited excellent biocompatibility and osteogenic properties. This study demonstrated the advantages of MBG in the preparation of Mg-based bone implants and validated the feasibility of improving Mg matrix corrosion resistance and enhancing osteogenesis by introducing MBG.

## 1. Introduction

Biodegradable magnesium (Mg) alloys as a revolutionary medical metal exhibit great potential for clinical application, in which they typically serve as screws, cardiovascular stents or implants [1,2,3]. For example, Shanghai Jiao Tong University has designed a new Mg-Nd-Zn-based alloy (JDBM), which has been prepared into bone plates, screws and even three-dimensional porous structures for bone tissue repair [4]. Mg alloys can gradually degrade in the human body until they are fully absorbed, cleverly meeting the clinical needs as a temporary substitute. The Mg ions generated by degradation take part in various enzyme syntheses, and can even promote bone healing. In addition, their Young’s modulus is close to that of human bone, which can effectively reduce the stress shielding effect [5,6,7,8,9,10]. However, their degradation is too fast in the bodily fluid environment, which leads to premature loss of mechanical structural integrity during service. Moreover, they will release a large amount of hydrogen gas, leading to subcutaneous swelling and local alkaline elevation [11,12,13].

Three main methods, including alloying, surface medication and Mg-based materials, have been proposed to tailor the corrosion rate of biomedical Mg alloys [14,15,16]. In particular, Mg-based composites use bioactive ceramics as calcium phosphate nucleation sites to induce surface mineralization and form a protective layer to delay corrosion. For instance, Wang X et al. [17] used a casting method to prepare a β-tricalcium phosphate (β- TCP)/Mg-Zn-Mn composite material, and its degradation performance was studied through immersion experiments. It was found that the calcium phosphate product layer deposited on the surface of the magnesium matrix delayed the degradation rate. Wan Y et al. [18] prepared bioglass/Mg composite materials using a microwave sintering method, and the results showed that the corrosion resistance, mechanical properties and biocompatibility were all improved to some extent.

In recent years, mesoporous bioglass (MBG) has attracted a lot of attention in the biomedicine field. MBG not only has the above excellent biological properties, but also has a uniform mesoporous structure (pore size of 2–50 nm) on the surface and a high specific surface area (500–800) [19,20,21]. Furthermore, MBG can hydrolyze in body fluids to form rich silanols and then polymerize into a negatively charged silica gel layer. The silica layer sequentially adsorbs Ca^2+^ and HPO_4_^2−^, thus promoting rapid surface mineralization and forming a dense calcium phosphorus layer. This structure serves as a protective layer to slow down the degradation of Mg alloys, as well as to improve their bioactivity and osteogenic capacity [22,23,24]. Due to its high specific surface area, MBG provides more adsorption sites than conventional bio-ceramic materials (e.g., β- TCP and BG) and thus is more favorable for the deposition of calcium phosphorus layers [25,26]. Notably, if nano-sized MBG is introduced into Mg alloys, it is usually uniformly distributed, stably embedded and cannot migrate easily in the matrix. Therefore, the in situ deposition of a calcium phosphorus layer occurs continuously to ensure space maintenance provision, thus also ensuring continuity of corrosion protection and osteogenesis [27].

In this study, MBG was synthesized via a modified sol–gel method and then introduced into pure Mg to form Mg-based composites via powder metallurgy technology to improve its corrosion resistance and biological properties. Therefore, the distribution state of MBG in the matrix was studied. The corrosion behavior was assessed using in vitro immersion and electrochemical tests. In addition, the biocompatibility and osteogenic activity were evaluated through cell experiments.

## 2. Materials and Methods

### 2.1. Preparation of MBG

MBG was prepared by a modified sol–gel method, as shown in Figure 1. In detail, cetyltrimethylammonium bromide (3.6 g) and trimethylamine (0.12 mL) were added into 100 mL of distilled water and stirred for 1 h at 50 °C. Subsequently, tetrahydrate calcium nitrate (3.39 g) was added to the above solution to obtain an aqueous solution. After mixing 5 mL of tetraethyl orthosilicate and 20 mL of cyclohexane by sonication, the mixture was slowly added to the obtained aqueous solution for 12 h under stirring. After centrifugation at 3000 rpm, a white precipitate was collected. Subsequently, the MBG was obtained by cleaning the white precipitate with distilled water, drying at 80 °C for 20 h and calcining at 650 °C for 3 h. All reagents were provided by Shanghai Aladdin Biochemical Technology Co., Ltd. (Shanghai, China).

The morphology and microstructure of the synthesized MBG were studied utilizing transmission electron microscopy (TEM, Tecnai G-20, FEI, Hillsboro, OR, USA) equipped with an energy dispersive spectrometer (EDS, Oxford Inca Energy 350, Oxford Instruments, Abingdon, UK) at 200 kV. The phase structure analysis of MBG was conducted using an X-ray diffractometer (XRD, Bruker, D8 Advance, Berlin, Germany) with Cu Kα radiation and at a step size of 5°/min. The surface area and pore size distribution were evaluated using the nitrogen adsorption–desorption technique at 77 K.

### 2.2. Samples Preparation and Microstructure Characterization

Spherical Mg powder (purity: 99.9%, particle size: 10–15 μm) was obtained from Tangshan Weihao Magnesium Powder Co., Ltd. (Qian’an, China). The preparation process of the sample is shown in Figure 1b. The Mg and MBG powders were mixed by a planetary ball mill at 200 rpm for 2 h. The mixed powders were pressed in cold press equipment at the compaction pressure of 45 MPa. After maintaining this pressure for 10 min, a green compact material with a diameter of 20 mm and a height of 2 mm was obtained. The green compact material was sintered using a solid-state sintering tube furnace (OTF-1200X-5L, Hefei Kejing Material Technology Co. Ltd., Hefei, China) at 590 °C for 2 h under an argon atmosphere.

The sintered part was mechanically polished using metallographic 800, 1000, 1500 and 2000 grit sandpaper. Then, it was washed with deionized water and dried. The microstructure of pure Mg and Mg/MBG samples was observed using a scanning electron microscope (SEM, EVO 18, ZEISS, Oberkochen, Germany) and optical microscopy (OM, DM4700, Leica, Wetzlar, Germany). Before observation, the polished surface was etched by nitric–alcohol solution for 10 s. In addition, the texture was investigated via an electron backscattering diffractometer (EBSD, Symmetry, Oxford Instruments, Oxford, UK). In detail, the samples were first electrolytically polished with acidic solution. Then, the EBSD image was captured at a step size of 1 µm. Channel 5 software was adopted to further achieve the inverse pole figure (IPF), where the principle is to project the measured diffraction intensity onto the equatorial projection plane using the polar beam red plane projection method.

### 2.3. Immersion Experiments

The as-sintered sample was immersed in simulated body fluid (SBF) solution (pH 7.4) for 1, 4 and 7 days at 37 °C. The SBF solution volume was 100 mL per square centimeter on the exposed surface of the sample. The generated hydrogen was collected continuously using a self-built hydrogen collection device and the corresponding volume value was recorded. The pH variation was monitored using a pH meter (FE28-Standard, Mettler Toledo Instruments Co. Ltd., Greifensee, Switzerland) every 12 h. During immersion, the concentrations of Ca and P ions were measured using an inductively coupled plasma atomic emission spectrometry technique (ICP-AES). After immersion for 1, 4 and 7 days, the sample was taken out of the SBF solution and washed using absolute alcohol. The corroded surface was observed using SEM, and the composition of corrosion products accumulated on the sample was analyzed using XRD. The corrosion product was removed by Cr_2_O_3_ and AgNO_3_ solution (200 g/L Cr_2_O_3_ and 10 g/L AgNO_3_), then the corrosion surface was observed via SEM.

### 2.4. Electrochemical Measurements

The electrochemical corrosion behavior was investigated in SBF by potentiodynamic polarization and electrochemical impedance spectroscopy (EIS). An electrochemical workstation (PARSTAT 4000 A, Princeton Applied Research, Oak Ridge, TN, USA) was used for the measurements, in which the sample, a platinum sheet and AgCl acted as the working electrode, counter electrode and reference electrode, respectively. Before the measurements, the sample was wrapped in epoxy resin, leaving an exposed surface of 1 cm^2^, then tested for 1500 s to achieve a steady state. Subsequently, a potentiodynamic polarization test was carried out at a sweep rate of 1 mV/s. The EIS test sample was pre-immersed in SBF solution for 72, 120 and 168 h to deposit a Ca-P layer. An EIS test was conducted with a range of 10–2 to 105 Hz and the result was fitted using ZSimpDemo 3.2 software.

### 2.5. Cell Cultivation

The extracts were collected though immersing the sample in complete Dulbecco modified Eagle medium (DMEM) at 37 °C for 3 days, in which the ratio of medium volume to exposure sample area was 0.8 mL/cm^2^. The bone marrow mesenchymal stem cells (BMSCs) were cultured in 96-well plates for 1 day. Then, the cells were cultured in extracts with culture medium which was refreshed daily. After culture for 1, 4 and 7 days, the cell morphology was captured using a fluorescence microscope after staining using calcium-AM for 1 h. In addition, a cell viability assessment was conducted using a CCK-8 assay. In detail, on day 1, 4 and 7, 10 μL of CCK-8 reagent was added to each well for 1 h of culture. The optical density was measured using a microplate reader (Rayto, Shenzhen, China) at 450 nm.

The alkaline phosphatase (ALP) activity of BMSCs was evaluated after rinsing using PBS solution. In detail, 4% paraformaldehyde was used to fix BMSCs for 15 min. The cells were stained using an ALP staining kit for 12 h and observed using a microscope. The quantitative ALP activities of BMSCs cultured in Mg and Mg/MBG extracts were evaluated using an ALP assay kit. The BMSCs were fixed using paraformaldehyde for 30 min after culturing for 14 and 21 days, respectively. Then, a PBS solution was used to rinse the cells and an alizarin red S staining solution was used to stain the cells for 20 min. The stained cells were visualized using a microscope.

### 2.6. Statistical Analysis

In the present study, three samples were prepared for each test, and they were repeated at least three times. GraphPad Prism 9.3 software was used to handle the experimental data, and the results are presented as means ± standard deviation, in which the significant difference was set at *p* < 0.05.

## 3. Results

### 3.1. Structural Characteristic of MBG

The morphology of self-synthesized MBG captured by TEM is shown in Figure 2a. The MBG nanoparticles exhibited a branch-like structure with an average particle size of ~40 nm. The lines of the branch-like structure within MBG were clear and ordered at high magnification, indicating a successful synthesis. The EDS analysis showed the distribution of O, Si, P and Ca elements. The XRD result of self-synthesized MBG was obtained and is shown in Figure 2b, exhibiting a broad diffraction peak at 20–30°. The peak is uniquely characteristic of amorphous SiO_2_ materials, and is consistent with previous studies [24]. The pore structure characteristics and surface area of the MBG were analyzed using the nitrogen adsorption–desorption method (Figure 2c). The MBG exhibited a typical IV isotherm pattern and hysteresis loops of type H1. The determined specific surface area for MBG was 656.45 m^2^/g. The pore distribution in the MBG was determined by analyzing N_2_ desorption branches using the BJH model in Figure 2d. The average pore diameter of the MBG was 18.09 nm.

### 3.2. Microstructure of Sintered Mg/MBG

The microstructure of pure Mg and Mg/MBG samples was examined using SEM and OM. In the case of the pure Mg sample, no visible precipitates were observed, as shown in Figure 3a. Moreover, the grains had an equiaxed shape with an average grain size of ~8.7 μm, as illustrated in Figure 3b. More importantly, the texture exhibited a preferred orientation with an inclination angle of approximately 45° from the surface normal. Regarding the Mg/MBG sample, the surface was studded with numerous fine nano particles that were uniformly distributed throughout the matrix, as shown in Figure 3c. EDS analysis (Figure 3d) confirmed that they were MBG nanoparticles. The boundary between the grains was well defined and clean, indicating a homogeneous microstructure (Figure 3e). The grains were uniformly distributed throughout the sample, with no discernible preferred orientation. Notably, the grain size in Mg/MBG was significantly smaller compared to that of the pure Mg part (Figure 3f). It is believed that introducing MBG particles into the Mg alloy inhibited the grain growth, thus resulting in fine grains [25].

### 3.3. Degradation Behavior

The degradation behavior was studied through immersion experiments for up to 7 days. The cumulative hydrogen evolution per 12 h is depicted in Figure 4a. During the first 24 h of immersion, the hydrogen volume increase rates of both pure Mg and Mg/MBG groups were similar. However, the increase rate of the Mg/MBG group decreased after 24 h. It was inferred that a substance with protective properties against the Mg matrix was formed during the degradation process, which protected the Mg matrix from further corrosion. The same degradation pattern was also presented after pH variation, as shown in Figure 4b. The pH for pure Mg groups increased continuously compared to that for Mg/MBG groups after about 12 h, whereas the increase rate of the latter significantly decreased. After immersion for 7 days, the pH of Mg/MBG groups was ~9.1, which was lower than that of the pure Mg group (~10.24). Therefore, the addition of MBG could improve the corrosion resistance of pure Mg.

The effects of variation in Ca and P ion concentrations in the SBF solution with various immersion periods are shown in Figure 4c,d. As for the Mg/MBG group, the Ca ion concentrations increased sharply in the first 24 hours, and then gradually decreased. A similar trend was also presented in P ion concentration variation, as shown in Figure 4d. In fact, the Ca and P ion concentrations increased in the initial stage due to the rapid hydrolysis of MBG in the Mg/MBG group. Subsequently, the concentrations gradually decreased, as the exposed MBG nanoparticles served as effective nucleation sites to consume them. With the ongoing deposition, a hydroxyapatite layer gradually formed on the matrix, which was able to protect the matrix from further corrosion. As for pure Mg samples, Ca and P ion concentrations reduced continuously during the immersion period, since the corrosion-produced layer could also induce the faint deposition of a Ca-P layer.

The corrosion morphologies after immersion in SBF solution for 1, 4 and 7 days were captured by SEM, as shown in Figure 5. It could be seen that the surfaces of both Mg and Mg/MBG were covered by corresponding corrosion products. Obviously, some dense white particles/clusters were coated on the matrix and they gathered tightly together as the immersion time increased. According to previous studies, the white agglomerations were considered to be apatite layers. As for the pure Mg sample, heavy corrosion products with noticeable cracks could be observed on the corrosion surface. It is known that the corrosion products formed on the pure Mg matrix are Mg(OH)_2_ with a loose porous structure. As the degradation progressed, Mg(OH)_2_ gradually peeled off, resulting in the corrosion morphology shown in Figure 5. The red dotted box was magnified, as shown in Figure 5.

An XRD analysis of the corrosion products was carried out, with the obtained results shown in Figure 6. As was expected, the corrosion products for Mg/MBG samples contained Mg(OH)_2_ and Ca_10_(PO_4_)_6_(OH)_2_. The peaks corresponding to Ca_10_(PO_4_)_6_(OH)_2_ were more dense than those corresponding to Mg(OH)_2_. In comparison, the corrosion products on the pure Mg part were mainly Mg(OH)_2_. It should be noted that strong diffraction peaks corresponding to the Mg phase at the diffraction angles of 34.34° and 36.69° were present in the XRD pattern of pure Mg corrosion products, while the relative intensities of the peaks in corrosion products from the Mg/MBG part were weaker. In fact, the corrosion product film on the surface of the Mg/MBG part was thicker and denser, which was hard to penetrate by X-ray. Thereby, relatively weak peaks of α-Mg were detected for the Mg/MBG part.

The surface morphologies of Mg and Mg/MBG after removing the corrosion products are shown in Figure 7. The matrixes of both Mg and Mg/MBG were subjected to continuous corrosion as the immersion time extended. After immersion for 7 days, a large number of cavities and deep pits formed on the matrix of pure Mg due to the continuous corrosion of the matrix by Cl- and the detachment of corrosion products. However, the matrix of the Mg/MBG part was subjected to relatively weak corrosion than the pure Mg part. This was because the MBG-induced Ca-P deposit on the matrix formed a protective film and effectively prevented the further corrosion by the invasive solution.

### 3.4. Electrochemical Behavior

A potentiodynamic polarization experiment was carried out, and the obtained polarization curves are depicted in Figure 8a. The corrosion potential (*E_corr_*) and current densities (*I_corr_*) were calculated via Tafel region extrapolation, as displayed in Table 1. The *E_corr_* of Mg/MBG was −1.36 ± 0.01 V, which was more positive than that of the pure Mg part (−1.47 ± 0.02 V). The *I_corr_* of the Mg/MBG sample was 12.48 ± 0.20 μA/cm^2^, and was lower than that of pure Mg (74.81 ± 0.70 μA/cm^2^). Both *E_corr_* and *I_corr_* confirmed that the corrosion resistance of Mg/MBG was higher than that of Mg. In the polarization curve, the anode branching of Mg/MBG was lower than that of pure Mg, whereas the cathode branching was similar. The results indicated that the dissolution of the Mg matrix was slowed down due to the formation of a Ca-P layer on the Mg/MBG matrix.

EIS tests were performed, and the impedance spectra evolution after 0, 72, 120 and 168 h of immersion in SBF solution is presented in Figure 8b. It was found that the EIS curve for Mg/MBG exhibited a capacitive loop and an inductive loop after immersion for 0 h, but exhibited only a capacitive loop after immersion for 72, 120 and 168 h. The inductance loop was related to pitting corrosion of the Mg matrix, which is not conducive to the corrosion resistance of Mg alloys [28,29]. It was noted that the size of the impedance loops significantly increased for Mg/MBG as the pre-soaking time was prolonged, which was related to the thickness evolution of the Ca-P protective layer. The evolution also could be verified through the phase angle θ dependence on frequency, as shown in Figure 8c. This indicated the growth of a Ca-P protective layer, evidenced by an increase in the impedance modulus in the frequency range of 10^−2^ to 10^5^.

The equivalent circuit fitted from the Nyquist plots is presented in Figure 8d. Typically, the circuit consisted of a resistor (*R_s_*) that represented the solution resistance, a capacitor (*Q_t_*) that represented the double-layer capacitance at the electrode–solution interface, and a charge transfer resistance (*R_t_*) that represented the rate-limiting step of the electrochemical reaction. Additionally, a corrosion product film impedance (*R_f_*) was also included to account for the protective effect of the corrosion film on the Mg/MBG sample. The detailed parameters of the equivalent circuit are listed in Table 2. Clearly, *Q_f_*, which is closely related to the thickness of the corrosion product film, tended to decrease. Generally, a decrease in the *Q_f_* value indicated an increase in corrosion film thickness. In addition, *R_t_* and *R_f_* progressively increased with the increase in pre-soaking time. This consistent tendency could result from an improved protective product film deposited on Mg/MBG, which proved that in situ deposition of apatite products enhanced the protective effect of the Mg alloy corrosion product layer [30,31,32].

### 3.5. Cell Behavior

An indirect method of live/dead cell staining was adopted to analysis the biocompatibility, and the live/dead cell staining images of BMSCs cultured for 1, 3 and 5 days are shown in Figure 9a. Overall, no obvious dead BMSCs (stained red) were observed in all the groups during the culture period. The number of BMSCs increased with the culture time, while the cell morphology changed from a round shape on day 1 to a fusiform shape on day 5. The BMSCs changed in number and morphology, reflecting the suitability of the culture medium containing the extract. On the same cultivation days, cell growth in the Mg/MBG groups was healthier than that of Mg groups.

A CCK-8 assay was used to determine the cell viability, and the results are presented in Figure 9b. A poor cellular activity was presented in 100% extracts due to high concentrations of ions. The cellular activity increased in 50% extract groups. In detail, the Mg/MBG group showed a cell viability of 69.8% after 1 day, 81.6% after 3 days and 90.1% after 5 days with 100% extracts, while those of the Mg group were 58.8%, 62.2% and 65.4%, respectively. As for 50% extracts, the cell viability increased to 94.5% for the Mg/MBG group and 69% for the Mg group, proving no cytotoxicity according to ISO standard 10,993–5 [33,34].

In addition to the CCK-8 assay, we also measured the alkaline phosphate (ALP) activity from the Mg/MBG and Mg (as a control) extracts, since ALP is a marker of early osteogenic differentiation and commonly used to assess the differentiation of BMSCs [35]. After 7 and 14 days of culture, more purple areas (ALP) were visible in the Mg/MBG group, as shown in Figure 10a, indicating that the Mg/MBG group exhibited a higher level of ALP activity as compared to pure Mg (*p* < 0.05 in all cases). The enhanced ALP activity was attributed to the doping of BMG that accelerated the dissolution of Ca and Si ions from the bioglass lattice structure, thus promoting the mineralization process and enhancing ALP expression. A quantitative analysis of ALP also further validated the above conclusion; the ALP expressions in the Mg/MBG group were ~2.3 and 2.4 times higher than those in the Mg group on days 7 and 14, respectively, as depicted in Figure 10b.

Calcium nodule deposition in the Mg/MBG and Mg groups was determined using the alizarin red staining (ARS) method, as displayed in Figure 10c. After 14 and 21 days of culture, calcium nodule (red) generation for the Mg/MBG group was higher than the Mg group, indicating that the addition of MBG nanoparticles enhanced osteogenic differentiation. Subsequently, the amount of ARS dye was measured and quantified on days 14 and 21 (Figure 10d); the amount of ARS dye increased with the incubation time, and higher levels of ARS dye were detected in the Mg-MBG group as compared to pure Mg, whether on day 14 or 21.

Based on the above results, MBG can provide favorable corrosion resistance and osteogenesis. However, the current study was only conducted to evaluate the in vitro corrosion and biological properties, and corresponding in vivo studies in animals have not yet been completed. Therefore, animal experiments for the MBG-containing Mg-based composite will be the focus of future research to further confirm MBG’s advantages and promote its development as a potential clinical material. According to the findings of Grau et al. [36], there was a significant increase in vascularization around the critical calvarial bone defects of mice due to the positive effects of Mg-based implanted scaffolds, although they could loosen due to rapid degradation. Therefore, an outlook can be presented that compounding MBG into Mg-based implants may be an effective way to improve the corrosion resistance to achieve better therapeutic outcomes of critical bone defects [37,38].

## 4. Conclusions

In this work, MBG with a size of ~40 nm was prepared using an improved sol–gel method, while Mg/MBG samples as potential bone substitutes were built by PM. Their corrosion resistance, biocompatibility and osteogenic differentiation ability were systematically studied. The specific conclusions are described as follows:(1)The prepared MBG nanoparticles presented as uniform and mesoporous with a branch-like structure. The specific surface area was extremely high at 656.45 m^2^/g, and the average pore diameter was 18.09 nm.(2)The distribution of MBG in the Mg matrix was uniform, which effectively suppressed the grain growth, thus resulting in fine grains. In addition, BMG worsened the texture of pure Mg, resulting in no obvious discernible preferred orientation.(3)The high specific surface area of MBG provided a large amount of attachment sites for Ca and P ion deposition, promoting the formation of an apatite protective layer. The layer was dense and effectively protected the Mg matrix from further corrosion.(4)Cell activity was enhanced due to the better corrosion resistance of Mg/MBG, as it provided a suitable environment for cell growth.

## Figures and Tables

**Figure 1 materials-16-06248-f001:**
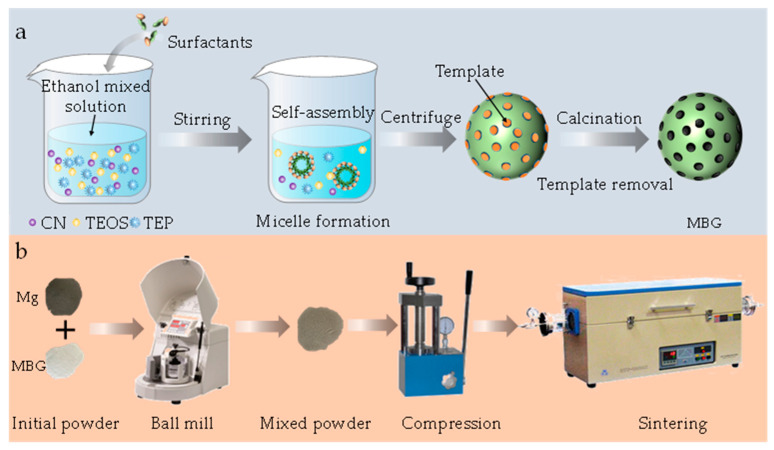
The synthesis process of MBG nanoparticles. (**a**) The preparation of the MBG (**b**) The preparation process of the sample.

**Figure 2 materials-16-06248-f002:**
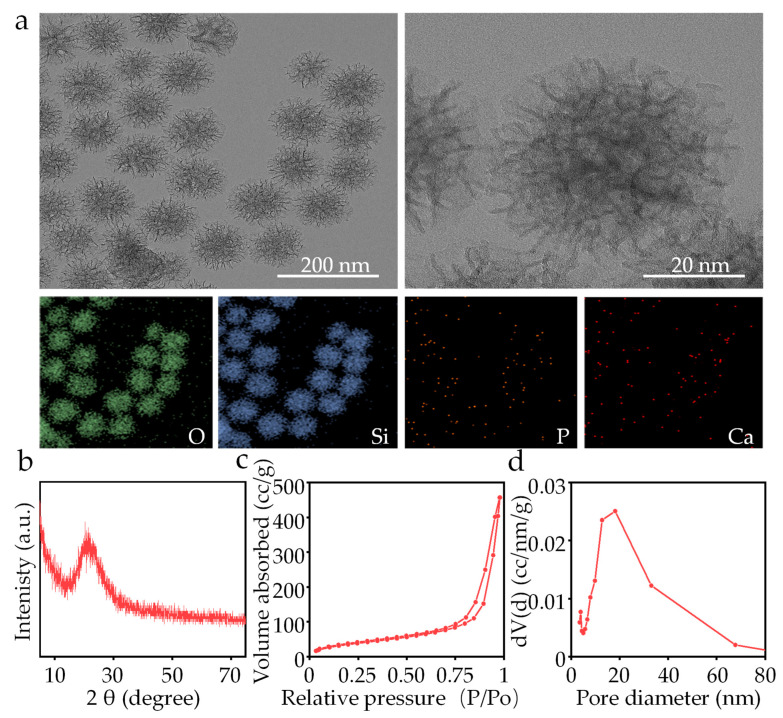
(**a**) TEM and EDS mapping of the MBG nanoparticles, (**b**) XRD spectrum, (**c**) BET isotherm and (**d**) pore size distribution of the MBG nanoparticles.

**Figure 3 materials-16-06248-f003:**
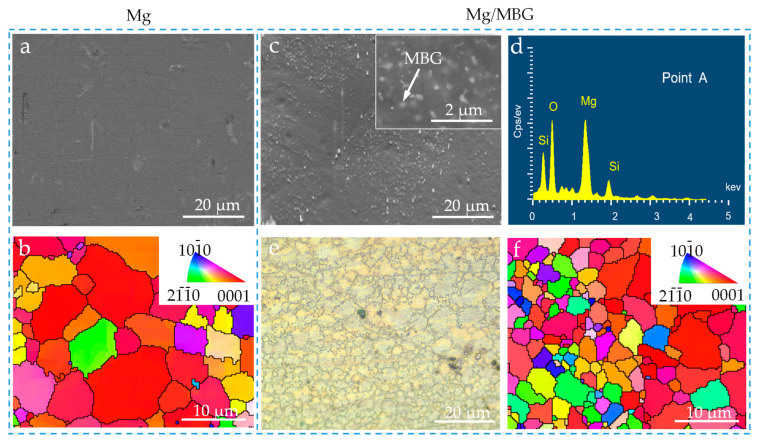
The microstructure of pure Mg and Mg/MBG samples. (**a**) SEM and (**b**) IPF maps of pure Mg sample; (**c**) SEM, (**d**) EDS analysis, (**e**) OM and (**f**) IPF maps of Mg/MBG sample. The insert in Figure 3c shows the uniform distribution of MBG in the Mg matrix, as marked by the arrows.

**Figure 4 materials-16-06248-f004:**
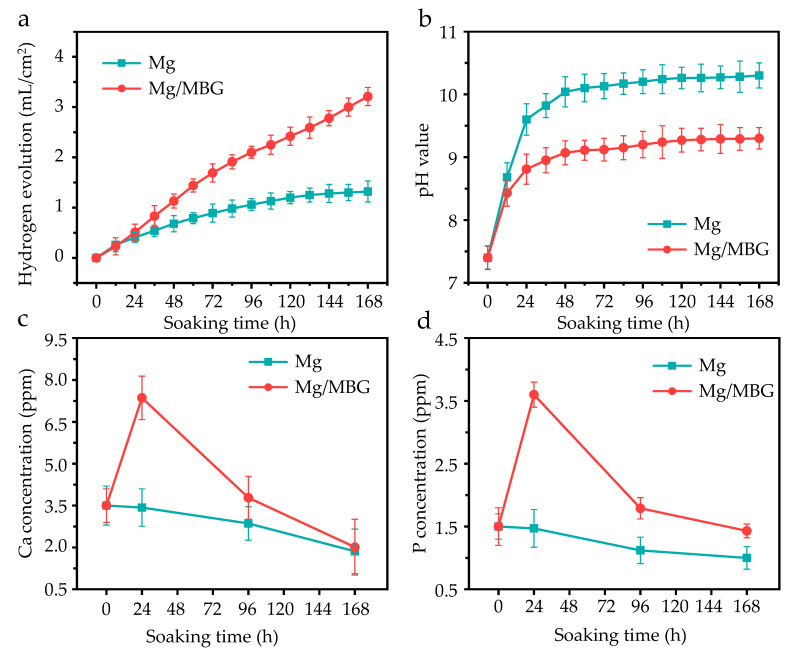
The degradation behavior of pure Mg and Mg/MBG in SBF for 7 days: (**a**) H_2_, (**b**) pH, (**c**) Ca ion and (**d**) P ion concentration variation.

**Figure 5 materials-16-06248-f005:**
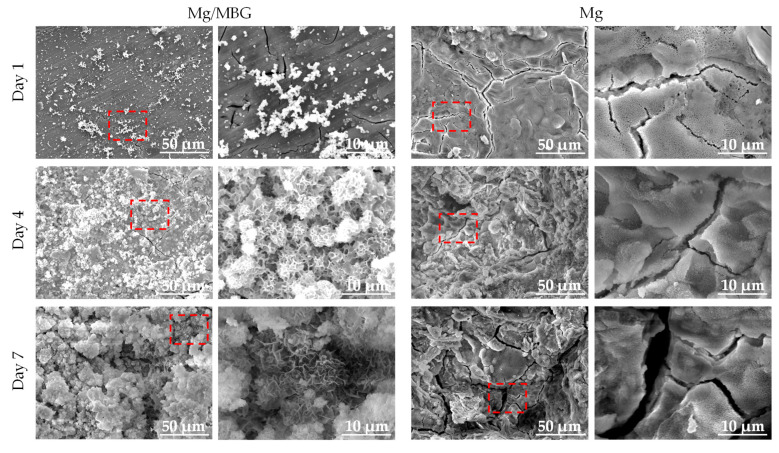
Corrosion morphology of Mg and Mg/MBG immersed in SBF solution for 1, 4 and 7 days.

**Figure 6 materials-16-06248-f006:**
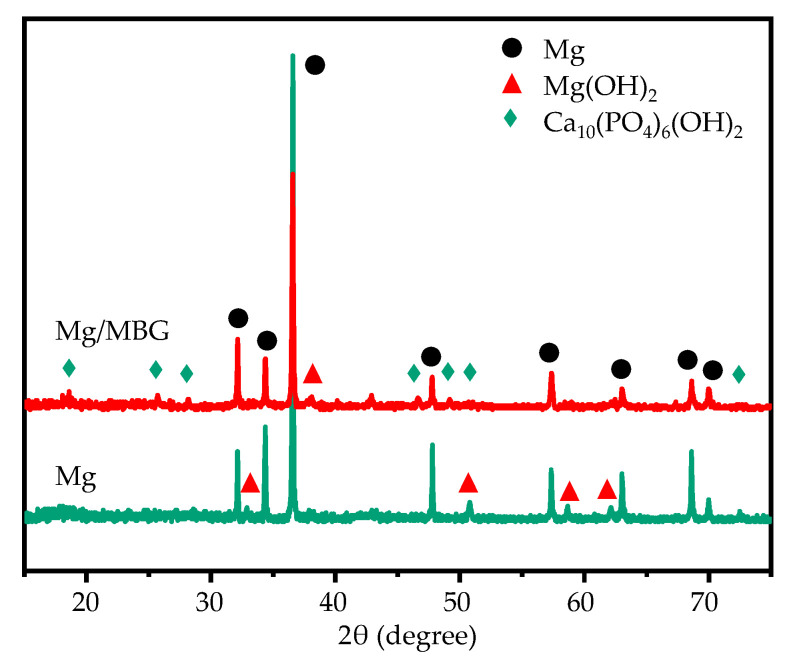
The XRD results for the corrosion products composition covered on Mg and Mg/MBG after immersion for 7 days.

**Figure 7 materials-16-06248-f007:**
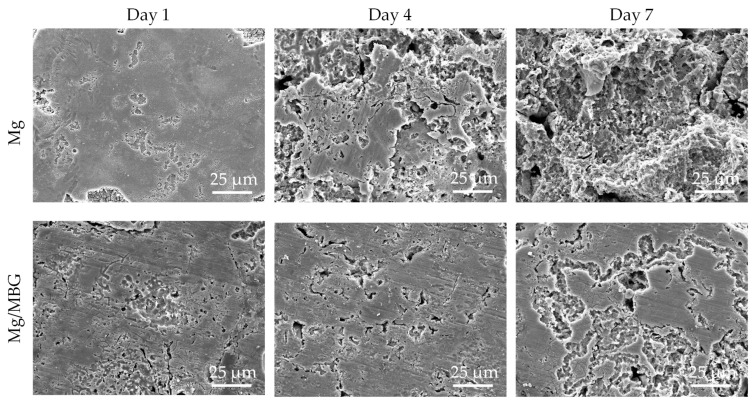
Corrosion morphology of pure Mg and Mg/MBG samples without corrosion product.

**Figure 8 materials-16-06248-f008:**
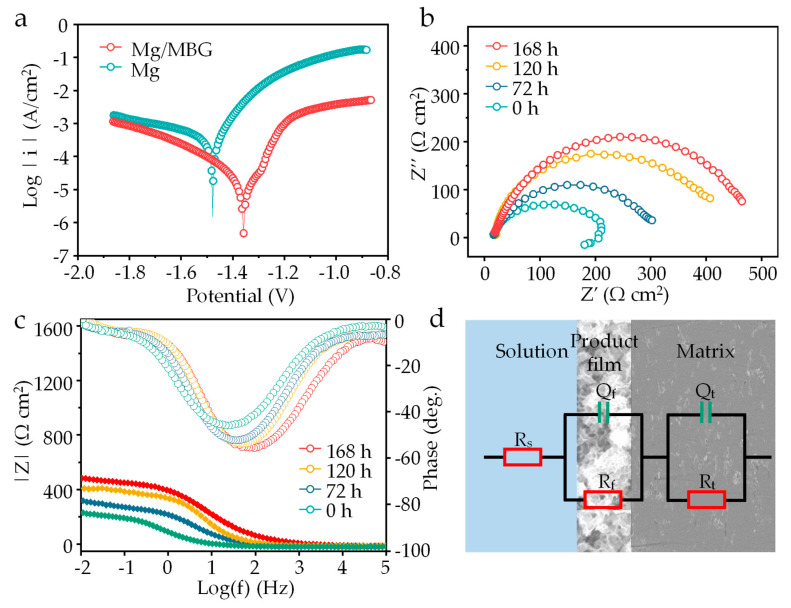
(**a**) Tafel polarization curves of Mg and Mg/MBG immersion for 0 h. (**b**) EIS plots, (**c**) Bode plots of Mg/MBG immersion for 0, 72, 120 and 168 h. (**d**) The equivalent electrical circuit.

**Figure 9 materials-16-06248-f009:**
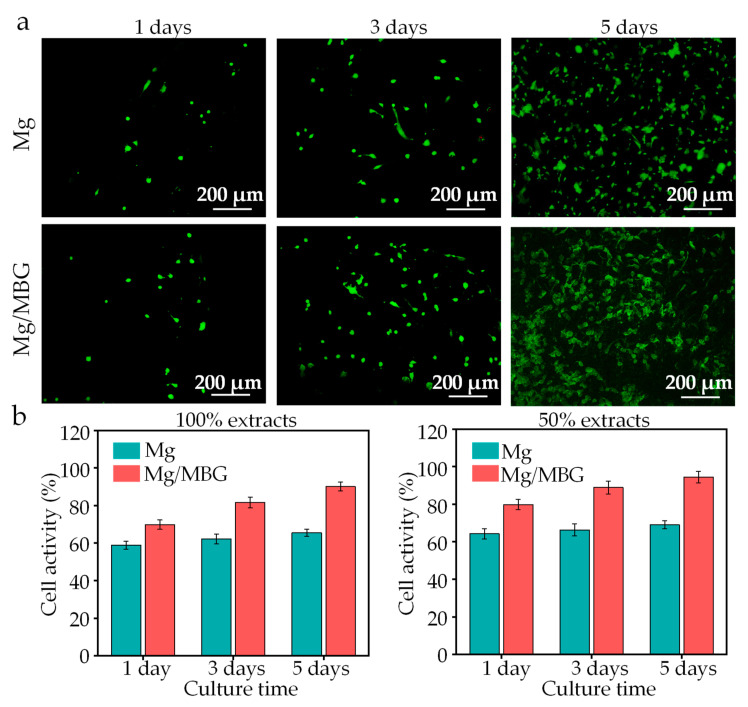
(**a**) Live/dead cell staining images of BMSCs; (**b**) cell viability of BMSCs for 100% and 50% extracts.

**Figure 10 materials-16-06248-f010:**
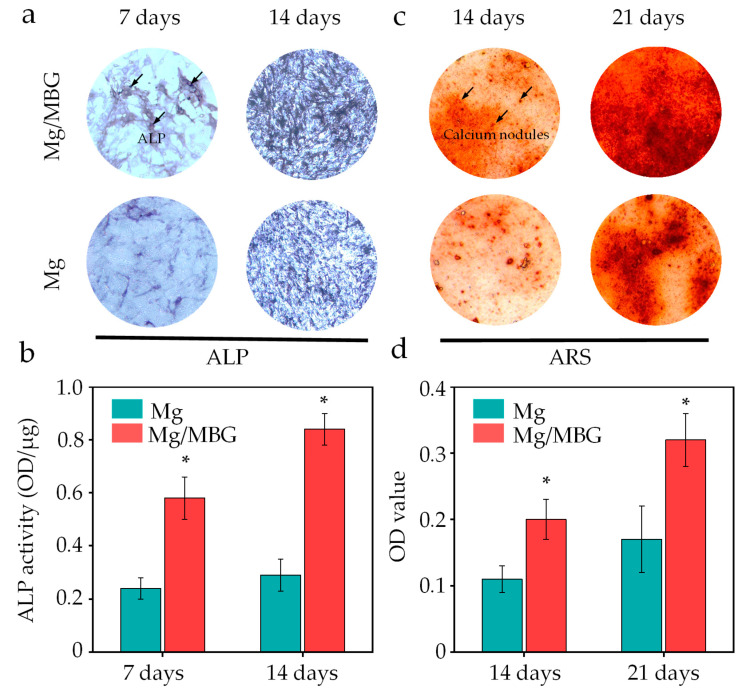
Osteogenesis experiments of BMSCs on Mg and Mg/MBG. (**a**) ALP staining on day 7 and 14, (**c**) Alizarin red staining on day 14 and 21 of BMSCs and (**b**,**d**) the corresponding quantitative analysis. * represent *p* < 0.05.

**Table 1 materials-16-06248-t001:** Corrosion potential and current densities of Mg and Mg/MBG calculated via Tafel region extrapolation.

Samples	*E_corr_* (V)	*I_corr_* (μA/cm^2^)
Mg	−1.47 ± 0.02	74.81 ± 0.70
Mg/MBG	−1.36 ± 0.01	12.48 ± 0.20

**Table 2 materials-16-06248-t002:** Fitted parameters of equivalent circuit for the Mg/MBG part after immersion.

Immersion Time (Hour)	*R_s_* (Ω·cm^2^)	*Q_t_* (S·sn·cm^−2^)	*R_t_* (Ω·cm^2^)	*R_f_* (Ω·cm^2^)	*Q_f_* (S·s^n^·cm^−2^)
0	9.76	6.36 × 10^−8^	184.3	58.47	1.24 × 10^−4^
72	9.21	6.95 × 10^−8^	248.52	169.72	9.37 × 10^−5^
120	10.03	7.26 × 10^−8^	373.86	259.64	8.24 × 10^−5^
168	9.83	7.38 × 10^−8^	418.36	374.22	5.63 × 10^−5^

## Data Availability

Not applicable.

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
