# Peer review of "Biodegradation and Cell Behavior of a Mg-Based Composite with Mesoporous Bioglass"

_materials, 2023, doi:10.3390/ma16186248_

Round 1
Reviewer 1 Report
Dear authors,
We have read with great interest your manuscript investigating MGP as bone filling material.
The overall project is very well presented, with conclusions supported by the clear results. All the results are convicing.
We have only few remarks :
- introduction : this section is a bit short, and lack of some explanation/exemples of potential applications of Mg materials (i.e in which form, for which specific application...)
- L.59: what is "telephone testing"??
- IPF is not defined in the manuscript, neither in the Methods section or Results. The principle of Inverse Pole Figure should be described in the Methods section.
- Figure 3: in addition to previous remark, the legend of Figure 3 should have more explanations.
- Conclusions : it would be of interest for the reader to get some potential clinical application, especially the way to use particles of 40nm size, as it is not obvious to see how such nanoparticles could be used as bone subtitute.
Reviewer 2 Report
MBG was synthesized via a modified sol-gel method and then introduced into the Mg matrix through powder metallurgy technology. The distribution state of MBG in the matrix was studied. The corrosion behaviour was studied using a physiological saline solution and telephone testing. In addition, biocompatibility and osteogenic activity were evaluated through cell experiments.
This research is under the scope of this journal; the topic is relevant for readers, and this research deals with potentially significant knowledge of the field. And It will be important for knowledge. The topic is relevant for readers and this review deals with potentially significant knowledge in the field and opens new ways for future studies.
However, there are some aspects that are possibly improved in the manuscript:
(Keywords)
- Please add more keywords, and order the keywords / Mesh terms alphabetically
(Introduction)
(Introduction)
- What is the importance of this review study? Which results are comparable with other articles? What has this study been new?
- Normally, Regeneration bone defects with scaffolds of the pores or the space provision versus compact materials, please read this article, (30. Palma, P. J., Matos, S., Ramos, J., Guerra, F., Figueiredo, M. H., & Krauser, J. (2010). New formulations for space provision and bone regeneration. Biodental Eng. I, 1, 71-76. WOS:000282776500012; SBN 978-0-415-57394-8.) reported the influence of different formulations of bone grafts in providing an adequate scaffold, thus emphasizing the importance of the three-dimensional distribution of particles and also space provision for new bone formation. It is very important to analyze the biodegradation of calcium phosphate Ceramics. What can be changed with this new material Mesoporous bioglass. The space provision sill importance for this materials…Add also in discussion.
(Materials &Methods)
- When mentioning materials or devices: please mention the manufacturer and city/ country.
- This section would better communicate to readers if restructured. A flowchart or diagram of the experimental procedure would be valuable. This section would better communicate to readers if restructured. A flowchart or diagram of the experimental procedure would be valuable.
- How many operators performed the Experimental Study? And how many times did you repeat the experimental study?
- Need to correct a ml for mL (mL is the international standard)
(Results)
- Improve the resolution quality of images, graphs/tables (and a presentation). The font/language in the figure/caption is different from the text. Please, standardise the size and the font in the figures with the font of the manuscript.
- On histological images, Identified the histological structures using arrows, letter
(Discussion)
- Please, identify the strength(s) and limitations of this study (as different canal conformation). And also, implications for future perspectives. Why will you use the implantation model instead of the Critical size defect on bone? The scaffolds are crucial for bone regeneration in critical-size defects. It seems crucial for biodegradation, For this reason, the authors need to support the necessity of using Scaffolds in “Critical size defect” by example animal studies. Please read these references. https://doi.org/10.3390/molecules26051339, https://doi.org/10.1111/j.1600-0501.2011.02179.x. (Explain this in the Discussion)
References)
- The titles of references have a different format,
The title of the article is written in capital letters at the beginning of words, others only in lowercase. Also, the standardised format of presentation in the journal's name. Because names are written in different formats, one is not abbreviated, others are not.
